# Event classification from the Urdu language text on social media

Malik Daler Ali Awan[1], Nadeem Iqbal Kajla[2], Amnah Firdous[3], Mujtaba Husnain[4] and Malik Muhammad Saad Missen[4]

[1] Department of Software Engineering, Faculty of Computing, The Islamia University of Bahawalpur, Bahawalpur, Punjab, Pakistan
[2] Department of Computer Science, Muhammad Nawaz Sharif University of Agriculture, Multan, Multan, Punjab, Pakistan
[3] Computer Science and Information Technology, The Govt. Sadiq College and Women University Bahawalpur, Bahawalpur, Punjab, Pakistan
[4] Department of Information Technology, Faculty of Computing, The Islamia University of Bahawalpur, Bahawalpur, Punjab, Pakistan



## ABSTRACT

The real-time availability of the Internet has engaged millions of users around the world. The usage of regional languages is being preferred for effective and ease of communication that is causing multilingual data on social networks and news channels. People share ideas, opinions, and events that are happening globally *i.e.*, sports, inflation, protest, explosion, and sexual assault, *etc.* in regional (local) languages on social media. Extraction and classification of events from multilingual data have become bottlenecks because of resource lacking. In this research paper, we presented the event classification task for the Urdu language text existing on social media and the news channels by using machine learning classifiers. The dataset contains more than 0.1 million (102,962) labeled instances of twelve (12) different types of events. The title, its length, and the last four words of a sentence are used as features to classify the events. The Term Frequency-Inverse Document Frequency (*tf-idf*) showed the best results as a feature vector to evaluate the performance of the six popular machine learning classifiers. Random Forest (RF) and K-Nearest Neighbor (KNN) are among the classifiers that out-performed among other classifiers by achieving 98.00% and 99.00% accuracy, respectively. The novelty lies in the fact that the features aforementioned are not applied, up to the best of our knowledge, in the event extraction of the text written in the Urdu language.

## INTRODUCTION

In the current digital and innovative era, text is still the strongest and dominant source of communication instead of pictures, emoji, sounds, and animations (*Lenhart et al., 2010*). The innovative environment of communication; real-time availability (*Motoyama et al., 2010*) of the Internet and unrestricted access for communication on social networks have attracted billions of people around the world. Now, people are hooked together *via* the Internet like a global village. They preferred to share detailed worthy information about

Corresponding author
Malik Daler Ali Awan,
daler.ali@iub.edu.pk

different topics, opinions, views, ideas, and events (*Reuter & Cimiano, 2012*) on social networks in different languages. The usage of different languages is being popular because social media and news channels have created space for local languages (*Rogstadius et al., 2013*). The Google input tool (https://www.google.com/inputtools/) provides language transliteration support for more than 88 different languages. Many other tools like the software (Inpage and Pak-Urdu for the Urdu language) provide the support to use local languages on social media for communication. Google Translate (https://translate.google.com/?hl=en) is a platform that facilitates multilingual users of more than 100 languages for conversation. Generally, people prefer to communicate in local languages instead of non-local languages for easiness.

A cursive language Urdu is one of the local languages that is being highly adapted for communication. There are more than 300 million (*Livingston, 2005*) Urdu language users all around the world that can speak, understand and write in the Urdu language. The Urdu language is a mix-composition of different languages *i.e.*, Arabic, Persian, Turkish, and Hindi (*Ghulam & Soomro, 2018*). In Pakistan and India, more than 65 million people can speak, understand, and write the Urdu language (*Naz & Hussain, 2013*). It is one of the resource-poor, neglected languages (*Mukund, Srihari & Peterson, 2010*) and the national language (*Nadeau & Sekine, 2007*) of Pakistan: the 6th most populous (https://www.worldometers.info/world-population/population-by-country/) country in the world. Urdu is also widely adopted and spoke as a second language all over the Pakistan (*Ghulam & Soomro, 2018*; *Naz & Hussain, 2013*; *Mukund, Srihari & Peterson, 2010*; *Nadeau & Sekine, 2007*).

In South Asia other countries (*Riaz, 2008*) *i.e.*, Bangladesh, Iran, and Afghanistan also have a considerable number of Urdu language users. Pak Urdu Installer (http://www.mbilalm.com/download/pak-urdu-installer.php) and Inpage are also common software, it support the Urdu language for textual writing (communication).

In contrast to cursive languages, there exists noteworthy work of information extraction and classification for *i.e.*, English, French, German, and many other non-cursive languages (*Nadeau & Sekine, 2007*; *Riaz, 2008*).

Sifting worthy insights from an immense amount of heterogeneous text existing on social media is an interesting and challenging task of Natural Language Processing (NLP). Event extraction and classification is one of the NLP tasks. The information of event classification is helpful to develop various NLP applications *i.e.*, to respond to emergencies, outbreaks, rain, flood, and earthquake (*Yin et al., 2012*), *etc*. Generally, people share their intent, appreciation, or criticism (*Purohit et al., 2015*) *i.e.*, enjoying discount offers by selling brands or criticizing the quality of the product. Earlier awareness of sentimental insights can be helpful to protect from business losses. The implementation of smart-cities possess a lot of challenges; decision making, event management, communication, and information retrieval. Extracting useful insights from an immense amount of text, dramatically enhance the worth of smart cities (*Alkhatibl, El Barachi & Shaalan, 2018*). Event information can be used to predict the effects of the event on the community, improve security and rescue the people.

Furthermore, classification of events can be used to collect relevant information about a specific topic, top-trends, stories, text summarization, and question and answering systems (*Khan et al., 2016*; *Jacobs, 1963*). Such information can be used to predict upcoming events, situations, and happening. For example, protesting events reported on social media generally end with conflict among different parties, injuries, death of people, and misuse of resources that cause anarchy. Some proactive measurements can be taken by the state to diffuse the situation and to prevent conflict. Similarly, event classification is crucial to monitor the law-and-order situation of the world.

Extracting and classification of event information from Urdu language text is a unique, interesting, and challenging task. The features of the Urdu language that made the event classification tasks more complex and challenging are listed below.

- Cursive nature of the script
- Morphologically enriched
- Different structures of grammar
- Right to the left writing style
- No text capitalization

Similarly, the lack of resources *i.e.*, the part-of-speech tagger (PoS), words stemmer, datasets, and word annotators are some other factors that made the processing of the Urdu text complex. There exist few noteworthy works related to the Urdu language text processing (see the literature for more details). All the above-mentioned factors motivated us to explore Urdu language text for our task.

## Concept of events

The definition of events varies from domain to domain. In literature, the event is defined in various aspects, such as a verb, adjective, and noun based depending on the environmental situation (*Ramesh & Suresh Kumar, 2016*; *Ahmed et al., 2016*). In our research work event can be defined as "An environmental change that occurs because of some reasons or actions for a specific period and influences the community." For example, the explosion of the gas container, a collision between vehicles, terrorist attacks, and rainfall, *etc.* There are several hurdles to process Urdu language text for event classification. Some of them are *i.e.*, determining the boundary of events in a sentence, identifying event triggers, and assigning an appropriate label.

## Event classification

"The automated way of assigning predefined labels of events to new instances by using pre-trained classification models is called event classification." Classification is supervised machine learning; all the classifiers are trained on label instances of the dataset.

## Multiclass event classification

It is the task of automatically assigning the most relevant one class from the given multiple classes. Some serious challenges of multiclassification are sentences overlapping in multiple classes (*Kong, Shi & Yu, 2011*; *Sarker & Gonzalez, 2015*) and imbalanced

instances of classes. These factors generally affect the overall performance of the classification system.

## Lack of resources

The researchers of cursive languages in the past were unexcited and vapid (*Mukund, Srihari & Peterson, 2010*) because of a lack of resources *i.e.*, dataset, part-of-speech tagger and word annotators, *etc.* Therefore, a very low amount of research work exists for cursive language *i.e.*, Arabic, Persian Hindi, and Urdu (*Pedregosa et al., 2011*). But now, from the last few years, cursive languages have attracted researchers. The main reason behind the attraction is that a large amount of cursive language data was being generated rapidly over the internet. Now, some processing tools also have been developed *i.e.*, part-of-speech taggers, word stemmers, and annotators that play an important role by making research handier. But these tools are still limited, commercial, and close domain.

Natural language processing is tightly coupled with resources *i.e.*, processing resources, datasets, semantical, syntactical, and contextual information. Textual features *i.e.*, Part of Speech (PoS) and semantic are important for text processing. Central Language of Engineering (CLE) (http://www.cle.org.pk/) provides limited access to PoS tagger because of the close domain and paid that diverged the researcher to explore Urdu text more easily.

Contextual features (*Vosoughi, Zhou & Roy, 2016*) *i.e.*, grammatical insight (tense), and sequence of words play important role in text processing. Because of the morphological richness nature of Urdu, a word can be used for a different purpose and convey different meanings depending on the context of contents. Unfortunately, the Urdu language is still lacking such tools that are publicly available for research. The dataset is the core element of research. The dataset for the Urdu language generally exists for name entity extraction with a small number of instances that are:

- Enabling Minority Language Engineering (EMILLE) (only 200,000 tokens) (*Baker et al., 2003*);
- Becker-Riaz *corpus* (only 50,000 tokens) (*Becker & Riaz, 2002*);
- International Joint Conference on Natural Language Processing (IJCNLP) workshop *corpus* (only 58,252 tokens);
- Computing Research Laboratory (CRL) annotated *corpus* (only 55,000 tokens are publicly available data corpora (*Malik, 2017*).

There is no specific dataset for events classification for Urdu language text.

## Concept of our system

The overall working process of our proposed framework is given in Fig. 1.

### Our contribution

- In this research article, we claim that we are the first ones who are exploring the Urdu language text to perform multi-class event classification at the sentence level using a machine learning approach,

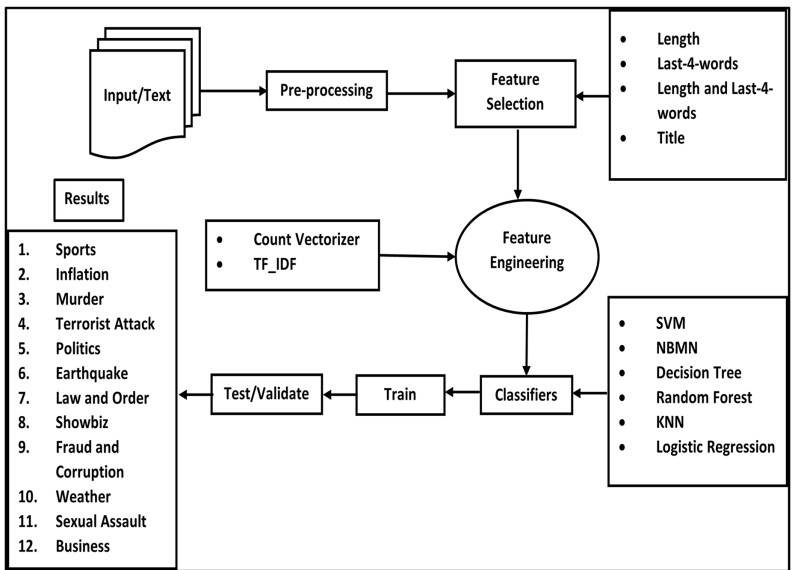

**Figure 1  Concept diagram.**

- A dataset that is larger than state-of-art used in experiments. In our best knowledge classification for twelve (12) different types of events never performed,
- A comprehensive and detailed comparison of six machine learning algorithms is presented to find a more accurate model for event classification for the Urdu language text.

### *Our limitations*

- There is no specific Word2Vec model for Urdu language text,
- There is also no availability of the free (open source) part-of-speech tagger and word stemmer for Urdu language text,
- Also, there exists no publicly available dataset of Urdu language text for sentence classification.

## RELATED WORK

Classification of events from the textual dataset is a very challenging and interesting task of Natural Language Processing (NLP). An intent mining system was developed (*Purohit et al., 2015*) to facilitate citizens and cooperative authorities using a bag of token model. The researchers explored the hybrid feature representation for binary classification and multi-label classification. It showed a 6–7% improvement in the top-down feature set processing approach. Intelligence information retrieval plays a vital role in the management of smart cities. Such information helps to enhance security and emergency management capabilities in smart cities (*Alkhatibl, El Barachi & Shaalan, 2018*). The textual content on social media is explored in different ways to extract event information. Generally, the event has been defined as a verb, noun, and adjective (*Nadeau & Sekine, 2007*). Event detection is a generic term that is further divided into event extraction and

event classification. A combined neural network of the convolutional and recurrent network was designed to extract events from English, Tamil, and Hindi languages. It showed 39.91%, 37.42% and 39.71% F_ Measure (*Ahmed et al., 2016*).

In the past, the researchers were impassive in cursive language, therefore a very limited amount of research work exists in cursive language *i.e.*, Arabic, Persian Hindi, and Urdu (*Alsaedi & Burnap, 2015*). Similarly, in the work of (*Alsaedi & Burnap, 2015*), the authors developed a multiple minimal reduct extraction algorithm which is an improved version of the Quick reduct algorithm (*Al-Radaideh & Al-Abrat, 2019*). The purpose of developing the algorithm is to produce a set of rules that assist in the classification of Urdu sentences. For evaluation purposes, an Arabic-based *corpus* containing more than 2,500 documents was plugged in for classifying them into one of the nine classes. In the experiment, we compared the results of the proposed approach when using multiple and single minimal reducts. The results showed that the proposed approach had achieved an accuracy of 94% when using multiple reducts, which outperformed the single reduct method which achieved an accuracy of 86%. The results of the experiments also showed that the proposed approach outperforms both the K-NN and J48 algorithms regarding classification accuracy using the dataset on hand.

Urdu textual contents were explored (*Daud, Khan & Che, 2017*) for classification using the majority voting algorithm. They categorized Urdu text into seven classes *i.e.*, Health, Business, Entertainment, Science, Culture, Sports, and Wired. They used 21,769 news documents for classification and reported 94% precision and recall. Dataset evaluated using these algorithms, Linear SGD, Bernoulli Naïve Bayes, Linear SVM, Naïve Bayes, random forest classifier, and Multinomial Naïve Bayes.

A framework (*Kuila, Chandra Bussa & Sarkar, 2018*) proposed a tweet classification system to rescue people looking for help in a disaster like a flood (*Singh et al., 2019*). The developed system was based on the Markov Model achieve 81% and 87% accuracy for classification and location detection, respectively. The features used in their system are (*Singh et al., 2019*):

- Number of words in a tweet (w);
- Verb in a tweet by (verb);
- Number of verbs in a tweet by (v);
- Position of the query by (Pos);
- Word before query word (before);
- Word after query word (after).

To classify Urdu news headlines (*Ali & Ijaz, 2009*) by using maximum indexes of vectors. They used stemmed and non-stemmed textual data for experiments. The system was specifically designed for text classification instead of event classification. The proposed system achieved 78.0% for competitors and 86.6% accuracy for the proposed methodology. In comparison, we used sentences of Urdu language for classification and explored the textual features of sentences. We have explored all the textual and numeric features *i.e.*, title, length, last-4-words, and the combinations of these (for more detail see Table 1)

**Table 1 Proposed features.**

| Sr. No. | Feature Name |
| --- | --- |
| 1 | Length |
| 2 | Last-4-words |
| 3 | Last-4-words and Length |
| 4 | Title |
| 5 | Title and Length |
| 6 | Title and Last-4-words |

in detail in this paper that were not reported ever in state-of-art according to our knowledge.

Twitter (*Agarwal & Rambow, 2010*) to detect natural disasters *i.e.*, bush fires, earthquakes and cyclones, and humanitarian crises (*Sakaki, Okazaki & Matsuo, 2010*). To be aware of emergencies situation in natural disasters a framework work designed based on SVM and Naïve Bayes classifiers using word unigram, bi-gram, length, number of #Hash tag, and reply. These features were selected on a sentence basis. SVM and Nave Bayes showed 87.5% and 86.2% accuracy respectively for tweet classification *i.e.*, seeking help, offering for help, and none. A very popular social website (Twitter) textual data was used (*Hussain, 2008*) to extract and classify events for the Arabic language. Implementation and testing of Support Vector Machine (SVM) and Polynomial Network (PN) algorithms showed promising results for tweet classification 89.2% and 92.7%. Stemmer with PN and SVM magnified the classification 93.9% and 91.7% respectively. Social events (*Usman et al., 2016*) were extracted assuming that to predict either parties or one of them aware of the event. The research aimed to find the relation between related events. Support Vector Machine (SVM) with kernel method was used on adopted annotated data of Automated Content Extraction (ACE). Structural information derived from the dependency tree and parsing tree is utilized to derive new structures that played important role in event identification and classification. The Tweet classification of the tweets related to the US airlines (*Rustam et al., 2019*) is performed by the sentiment analysis companies that are not related to our work. We tried to classify events at sentence level that is challenging since the Urdu sentence contains very short features as compared to a tweet. It is pertinent to mention that the sentiment classification is different from the event classification. Multiclass event classification is reported (*Ali, Missen & Husnain, 2021*) comprehensively, deep learning classifiers are used to classify events into different classes.

## MATERIALS & METHODS

Event classification for Urdu text is performed using a supervised machine learning approach. A complete overview of the multi-class event classification methodology is given in *Fig. 1*. Textual data classification possesses a lot of challenges *i.e.*, word similarity, poor grammatical structure, misuse of terms, and multilingual words. That is the reason, we

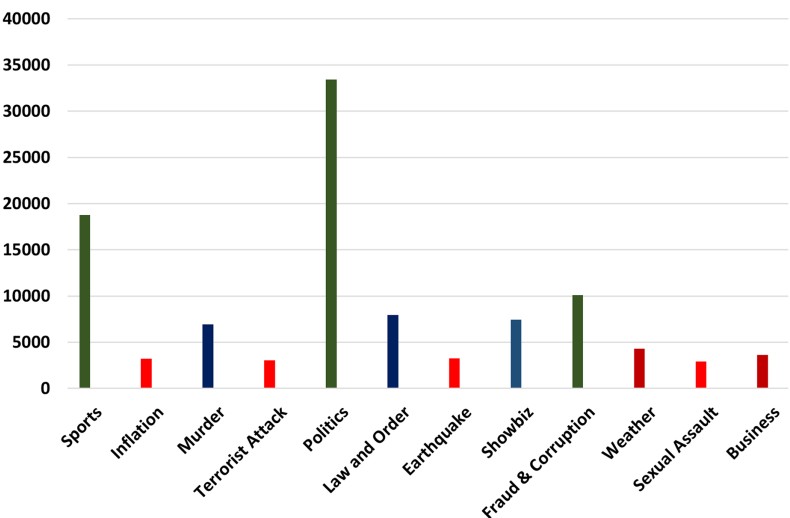

**Figure 2** **Maximum number of instances.**

decided to adopt a supervised classification approach to classify Urdu sentences into different categories.

## Data collection

Urdu data were collected from popular social networks (Twitter), famous news channel blogs *i.e.*, Geo News (https://urdu.geo.tv/), Urdu Point (https://www.urdupoint.com/daily/), and BBC Urdu (https://www.bbc.com/urdu). The data collection consists of the title, the main body, the published date, the location, and the URL of the post. In the phase of data collection, a PHP-based web scraper is used to crawl data from the above-mentioned social websites. A complete post is retrieved from the websites and stored in MariaDB (database). Our dataset consists of 0.1 million (102,960) label sentences of different types of events. All the different types of events used in our research work and their maximum number of instances are shown below in Fig. 2.

There are twelve different types of events that we try to classify in our research work. These events are a factual representation of the state and the situation of the people. In Fig. 2 imbalances number of instances of each event are given. It can be visualized that politics, sports, and Fraud & Corruption have a higher number of instances while Inflation, Sexual Assault, and Terrorist attacks have a lower number of instances. These imbalanced numbers of instances made our classification more interesting and challenging.

Multiclass events classification tasks are comprised of many classes. The different types of events that are used in our research work *i.e.*, sports, Inflation, Murder & Death, Terrorist attacks, Politics, Law and Order, Earthquake, Showbiz, Fraud & Corruption, Weather, Sexual Assault, and Business. All the sentences of the dataset are labeled by the above-mentioned twelve (12) different types of events. Finally, a numeric (integer) value is assigned to each type of event label (See Table 2 for more details of the label and its relevant numeric value).

**Table 2 Types of events and their labels in the dataset.**

| Event | Label | Event | Label |
|---|---|---|---|
| Sports | 1 | Earthquake | 7 |
| Inflation | 2 | Showbiz | 8 |
| Murder and Death | 3 | Fraud and Corruption | 9 |
| Terrorist Attack | 4 | Rain/Weather | 10 |
| Politics | 5 | Sexual Assault | 11 |
| Law and Order | 6 | Business | 12 |

## Preprocessing

The initial preprocessing steps are performed on the *corpus* to prepare it for machine learning algorithms. Because textual data cannot directly process by machine learning classifiers. It also contains many irrelevant words. The detail of all the preprocessing steps is given below. These steps were implemented in a PHP-based environment, while the words tokenization is performed using the scikit library (*Pedregosa et al., 2011*) in Python.

## Post splitting

The PHP crawler extracted the body of the post. It comprises many sentences as a paragraph. In the Urdu language script, sentences end with a sign called "-" Hyphen (Khatma-ﺘﻤہ). It is a standard punctuation mark in the Urdu language to represent the end of the sentence. As mentioned earlier, we are performing event classification at the sentence level. So, we split paragraphs of every post into sentences. Every line in the paragraphs ending at Hyphen is split as a single line.

## Stop words elimination

Generally, those words that occur frequently in text *corpus* are considered as stop words. These words merely affect the performance of the classifier. Punctuation marks ("!", "@", "#", *etc.*) and frequent words of the Urdu languages (کا (ka), کے (kay), کی (ki), *etc.*) are the common examples of stop words. All the stop words (*Kuila, Chandra Bussa & Sarkar, 2018*) that do not play an influential role in event classification for the Urdu language text are eliminated from the *corpus*. Stop words elimination reduces memory and processing utilization and makes the processing efficient.

## Noise removal and sentences filtering

Our data were collected from different sources (see "Materials & Methods"). It contains a lot of noisy elements *i.e.*, multilanguage words, links, mathematical characters, and special symbols, *etc.* To clean the *corpus*, we removed noise *i.e.*, multilingual sentences, irrelevant links, and special characters.

The nature of our problem confined us to define the limit of words per sentence. Because of the multiple types of events, it is probably hard to find a sentence of the same length. We decided to keep the maximum number of sentences in our *corpus*. All those sentences which are brief and extensive are removed from our *corpus*. In our dataset lot of sentences varying in length from five words to 250 words. We decided to use sentences that

**Table 3 Last four words representing an event.**

| Event | | Non_ event | |
|---|---|---|---|
| **Urdu** | **English** | **Urdu** | **English** |
| مسئلہ کشمیر کو لے کر پاکستان اور بھارت میں جنگ چھڑ چکی ہے۔ | The battle between Pakistan and India has been started on the conflict of Kashmir. | چند دن پہلے لوگ خوش تھے۔ | Few days ago, people were happy. |

consist of five words to 150 words to lemmatize our research problem and to reduce the consumption of processing resources.

## Sentence labeling

In supervised learning, providing output (Label) detail in the *corpus* is a core element. Sentence labeling is an exhausting task that requires deep knowledge and an expert's skill of language. All the sentences were manually labeled by observing the title of the post and body of sentences by Urdu language experts (see Table 2 for sentence labeling). Three Urdu language experts were engaged in the task of sentence labeling. One of them is Ph.D. (Scholar) while the other two are M.Phil. To our best knowledge, it is the first largest labeled dataset for the multi-class event in the Urdu language.

## Feature selection

The performance of prediction or classification models is cohesively related to the selection of appropriate features. In our dataset six (6) features excluding "Date" as a feature are considered valuable to classify Urdu news sentences into different classes. All the proposed features that are used in our research work are listed in Table 1.

## Why were these features selected?
### Last-4-words of sentence
Occurrence, happening, and situations are generic terms that are used to represent events. In general, "verb" represents an event. The grammatical structure of Urdu language is Subject_ Object_ Verb (SOV) (*Agarwal & Rambow, 2010*), which depicts that verb, is laying in the last part of the sentences.

For example, the sentence ("احمد نے پودوں کو پانی دیا۔" – Ahmad ney podon ko pani dia"), (Ahmad watered the plants) follows the SOV format. "Pani dia-پانی دیا" is the verbal part of the sentence existing in the last two words of the sentence. It shows the happening or action of the event. Our research problem is to classify sentences into different classes of events. So, that last-4-words are considered one of the vital features to identify events and non-event sentences. For example, in Table 3 in the event column underline/highlighted part of the sentence represents the happening of an event *i.e.*, last-4-words in the sentence, while labeling the sentences we are strictly concerned that only event sentences of different types should be labeled.

### Title of post
Every conversation has a central point *i.e.*, title. Textual, pictorial, or multimedia content that is posted on social networks as a blog post, at the paragraph level or sentence level

describes the specific event. Although many posts contain irrelevant titles to the body of the message. However, using the title as a feature to classify sentences is crucial because the title is assigned to the contents-based material.

### Length of sentence

A sentence is a composition of many words. The length of the sentence is determined by the total number of words or tokens that exist in it. It can be used as a feature to classify sentences because many sentences of the same event have probably the same length.

### Title and length

The proposed feature is the combination of the title of the post and the length of the sentence. The title represents the central idea of the post, and the length of the sentence varies from title to title.

### Title and last-4-words

The combination of title and last-4-words in Urdu language text is very helpful for classifying the sentences, because last-4-words generally represent the occurrence/happening of some event.

### Length and last-4-words

We also consider the combination of length with last-4-words as a valuable feature because the length of a sentence varies from event to event.

### Features engineering

Feature Engineering is a way of generating specific features from a given set of features and converting selected features to machine-understandable format. Our dataset is text-based that consists of more than 1 million (102,960 labeled) instances *i.e.*, sports, inflation, death, terrorist attack, and sexual assault, *etc*. 12 classes.

As mentioned earlier that the Urdu language is one of the resource-poor languages and since there are no pre-trained word embedding models to generate the embedding vectors for Urdu language text, we could not use the facility of Word2Vec embedding technique.

All the textual features are converted to numeric format *i.e.* (Term Frequency_ Inverse Document Frequency) $TF_{IDF}$ and Count-Vectorizer. These two features $TF_{IDF}$ and Count-Vectorizer are used in a parallel fashion. The scikit-learn package is used to transform text data into numerical value (*Pedregosa et al., 2011*).

### Count_ vectorization

The process of converting words to numerical form is called vectorization. Its working strategy is based on term frequency. It counts the frequency of specific word w and builds the spare matrix-vector using bag-of-words (BOW). The length of the feature vector depends on the size of the bag-of-words *i.e.*, dictionary.

### Term frequency inverse document frequency

It is a statistical measure of word w to understand the importance of that word for specific document d in the *corpus*. The importance of a word is proportionally related to frequency

*i.e.*, higher frequency more important. The mathematical formulas related to $\text{TF}_{\text{IDF}}$ are given below:

$$\text{Term Frequency (TF)} = \frac{\textit{Number of time term t appears in document}}{\textit{Total number of terms in documents}} \tag{1}$$

$$\text{Inverse Document Frequency (IDF)} = Log_e \frac{\textit{Total number of document}}{\textit{Total number of documents term t appears}} \tag{2}$$

$$\text{TF}_{\text{IDF}} = TF * IDF \tag{3}$$

# EXPERIMENTAL SETUP

Classifiers are the algorithms used to classify data instances into predefined categories. Many classifiers exist that process the textual data using a machine learning approach. In our research work, we selected the six most popular machine learning algorithms *i.e.*, Random Forest (RF) (*Livingston, 2005*), K-Nearest Neighbor (KNN), Support Vector Machine (SVM), Decision Tree (DT), Naïve Bayes Multinomial (NBM), and Linear Regression (LR).

## Machine learning classifiers

In this section, we presented the detail of six classifiers that were used to classify the Urdu sentences using different proposed features.

### Random forest (RF)

This model is comprised of several decision trees that act as a building block of RF. Every decision tree is created using the rules *i.e.*, if then else, and the conditional statements, *etc.* (*Livingston, 2005*). These rules are then followed by the multiple decision trees to analyze the problem at a discrete level.

### k-Nearest neighbor

It is one of the statistical models that find the similarity among the data points using Euclidean distance (*Guo et al., 2003*). It belongs to the category of lazy classifiers and is widely used for classification and regression tasks.

### Support vector machine

It is based on statistical theory (*Zhang, 2012*), to draw a hyperplane among points of the dataset. It is highly recommended for regression and classification *i.e.*, binary classification, multiclass classification, and multilabel classification. It finds the decision boundary to identify different classes and maximize the margin.

### Decision tree

It is one of the supervised classifiers that work following certain rules. Data points/inputs are split according to the specific condition (*Zhong, 2016*). It is used for regression and classification using the non-parametric method because it can handle textual and numerical data. Learning from data points is accomplished by approximating the sine

curve with the combination of an if-else-like set of rules. The accuracy of a model is related to the deepness and complexity of rules.

### Naïve Bayes multinominal

It is a computationally efficient classifier for text classification using discrete features. It can also handle the textual data by converting it into numerical (*Xu, 2018*) format using count vectorizer and term frequency-inverse document frequency (tf-idf).

### Linear regression

It is a highly recommended classifier for numerical output. It is used to perform prediction by learning linear relationships between independent variables (inputs) and dependent variables (output) (*Zhang & Oles, 2001*).

### Training dataset

A subpart of the dataset that is used to train the models to learn the relationship among dependent and independent variables is called the training dataset. We divided our data into training and testing using the train_ test_ split function of the scikit library using python. Our training dataset consists of 70% of the dataset that is more than 70,000 labeled sentences of Urdu language text.

### Testing dataset

It is also the subpart of the dataset that is usually smaller than size as compared to the training dataset. In our research case, we decided to use 30% of the dataset for testing and validating the performance of classifiers. It comprises more than 30,000 instances/sentences of Urdu language text.

### Performance measuring parameters

The most common performance measuring parameters (*Nadeau & Sekine, 2007*; *Riaz, 2008*; *Ramesh & Suresh Kumar, 2016*; *Ahmed et al., 2016*; *Kong, Shi & Yu, 2011*) *i.e.*, precision, recall, and F1_measure are used to evaluate the proposed framework since these parameters are the key indicators while performing the classification in a multiclass environment using an imbalanced dataset.

$$\text{Precision} = \frac{TP}{TP + FP} \qquad (4)$$

$$\text{Recall} = \frac{TP}{TP + FN} \qquad (5)$$

$$F1 = 2 * \frac{\text{Precision} * \text{Recall}}{\text{Precision} + \text{Recall}} \qquad (6)$$

$$\text{Accuracy} = \frac{TP + TN}{TP + TN + FP + FN} \qquad (7)$$

## RESULTS

To evaluate our dataset, the Python package scikit-learn is used to perform event classification at the sentence level. We extracted the last four words of each sentence and

**Table 4 Length.**

| Algorithms | Accuracy (%) | Feature |
|---|---|---|
| SVM | 17 | Length |
| NBM | 32 | |
| LR | 32 | |
| Decision Tree | 32 | |
| Random Forest | 32 | |
| K-NN | 24 | |

**Table 5 Last _4_words accuracy.**

| Algorithms | Accuracy (%) | Feature |
|---|---|---|
| SVM | 45 | Last _4_words |
| NBMN | 44 | |
| LR | 49 | |
| Decision Tree | 49 | |
| **Random Forest** | **52** | |
| K-NN | 48 | |

**Table 6 Last _4_words and length accuracy.**

| Algorithms | Accuracy (%) | Feature |
|---|---|---|
| SVM | 46 | Length and Last _4_words |
| NBMN | 44 | |
| LR | 49 | |
| Decision Tree | 48 | |
| **Random Forest** | **53** | |
| K-NN | 49 | |

calculated the length of each sentence. To obtain the best classification results we evaluated six machine learning classifiers among others *i.e.*, Decision Tree (DT), Random Forest (RF), Logistic Regression (LR), Support Vector Machine (SVM), *k*-Nearest Neighbor, and Naïve Bayes Multinominal (NBM).

We proposed three features *i.e.*, Length, Last-4-words, and Length and Last-4-words to classify sentences into different types of events (see Table 2). The results were obtained using 'length' as the feature is shown in Table 4. The classifiers *i.e.*, DT, RF, NBM, and LR showed 32% accuracies that is very low. The comparatively second feature that is Last-4-words showed better results for these above-mentioned classifiers. Random Forest showed 52% accuracy that is a considerable result as an initiative for multiclass event classification in the Urdu language text. The detail of results regarding other classifiers can be seen in Table 5.

We also evaluated these classifiers using another feature that is the combination of both Length and Last-4-grams. It also improved the overall 1% accuracy of the proposed system. The Random Forest showed 53.00% accuracy. The further details of accuracies of other used machine learning models can be seen in Table 6.

**Table 7 Title and Last _4_words accuracy.**

| Algorithms | Accuracy (%) | Feature |
|---|---|---|
| SVM | 85 | Title and Last _4_words |
| NBMN | 91 | |
| LR | 95 | |
| Decision Tree | 97 | |
| **Random Forest** | **98** | |
| **K-NN** | **99** | |

**Table 8 Title and length.**

| Algorithms | Accuracy (%) | Feature |
|---|---|---|
| SVM | 87 | Title and Length |
| NBMN | 93 | |
| LR | 98 | |
| **Decision Tree** | **99** | |
| **Random Forest** | **99** | |
| K-NN | 94 | |

**Table 9 Random forest TP, FN, FP and TN.**

| | Random forest | | | | |
|---|---|---|---|---|---|
| Label | Type of event | TP | FN | FP | TN |
| 1 | Sports | 5,646 | 15 | 14 | 25,514 |
| 2 | Inflation | 967 | 0.0 | 08 | 30,211 |
| 3 | Murder and Death | 2,096 | 19 | 22 | 29,052 |
| 4 | Terrorist Attack | 865 | 13 | 06 | 30,304 |
| 5 | Politics | 9,983 | 47 | 86 | 21,073 |
| 6 | law and order | 2,257 | 36 | 23 | 28,872 |
| 7 | Earthquake | 970 | 0.0 | 0.0 | 30,219 |
| 8 | Showbiz | 2,244 | 15 | 04 | 28,929 |
| 9 | Fraud and corruption | 3,015 | 29 | 21 | 35,924 |
| 10 | Rain/weather | 1,031 | 0.0 | 05 | 34,888 |
| 11 | Sexual Assault | 889 | 0.0 | 01 | 30,300 |
| 12 | Business | 1,032 | 20 | 04 | 30,134 |

The results obtained by using the above features are very low, we deiced to use the title of the post as a feature to improve the performance of the system. We integrated the "Title" of the post with each sentence of the same paragraph that dramatically improves the accuracy of the system. We combined the "Title" of the post with other features *i.e.*, length, and Last-4-words. The detail of the highest accuracies that is obtained by the combination of these features *i.e.*, Last-4-words, length, and title are given in Tables 7 and 8. Random forest and *k*-NN showed the highest accuracies. The detail of the confusion matrix related to the proposed system (TP, FP, TN, FN) is also given in Tables 9 and 10.

**Table 10 KNN TP, FN, FP and TN.**

| | K-nearest neighbor | | | | |
|---|---|---|---|---|---|
| Label | Type of event | TP | FN | FP | TN |
| 1 | Sports | 5,638 | 23 | 34 | 25,494 |
| 2 | Inflation | 967 | 0.0 | 29 | 30,139 |
| 3 | Murder and Death | 2,077 | 38 | 32 | 29,044 |
| 4 | Terrorist Attack | 858 | 20 | 21 | 30,308 |
| 5 | Politics | 9,931 | 99 | 98 | 21,052 |
| 6 | law and order | 2,238 | 55 | 42 | 28,854 |
| 7 | Earthquake | 970 | 0.0 | 07 | 30,219 |
| 8 | Showbiz | 2,242 | 17 | 21 | 28,908 |
| 9 | Fraud and corruption | 3,023 | 21 | 13 | 28,121 |
| 10 | Rain/weather | 1,031 | 0.0 | 26 | 10,145 |
| 11 | Sexual Assault | 889 | 0.0 | 11 | 30,293 |
| 12 | Business | 1,001 | 51 | 04 | 30,133 |

**Table 11 Random forest performance using the title, and last _4_words.**

| Label | Event | Precision | Recall | F1_Measure |
|---|---|---|---|---|
| 1 | Sports | 0.99 | 0.99 | 0.99 |
| 2 | Inflation | 0.99 | 1.00 | 0.99 |
| 3 | Murder and Death | 0.98 | 0.99 | 0.98 |
| 4 | Terrorist Attack | 0.97 | 0.96 | 0.97 |
| 5 | Politics | 0.98 | 0.99 | 0.98 |
| 6 | law and order | 0.98 | 0.96 | 0.97 |
| 7 | Earthquake | 1.00 | 1.00 | 1.00 |
| 8 | Showbiz | 0.99 | 0.98 | 0.99 |
| 9 | Fraud and corruption | 0.99 | 0.98 | 0.98 |
| 10 | Rain/weather | 1.00 | 1.00 | 1.00 |
| 11 | Sexual Assault/Intercourse | 1.00 | 1.00 | 1.00 |
| 12 | Business | 0.98 | 0.95 | 0.97 |
| Overall accuracy | 98.53% | | | |

The standard performance measuring parameters *i.e.*, precision, recall, and f1-measure of Random Forest and *k*-NN classifiers using "Title and Last-4words" as features are given in Tables 11 and 12 respectively. Similarly other combinations of features *i.e.*, "Title and Length" are used to enhance the accuracy of the system. The Decision Tree and Random Forest showed the highest results as compared to other classifiers for this specific combination of features. A detailed summary of the results related to Decision Tree and Random Forest is given in Tables 13 and 14 respectively.

We finally presented the comparison of four classifiers that showed the highest results in Fig. 3.

**Table 12 KNN performance using the title, and last _4_words.**

| Label | Event | Precision | Recall | F1_Measure |
|---|---|---|---|---|
| 1 | Sports | 0.99 | 1.00 | 0.99 |
| 2 | Inflation | 0.97 | 1.00 | 0.99 |
| 3 | Murder and Death | 0.99 | 0.98 | 0.98 |
| 4 | Terrorist Attack | 0.98 | 0.98 | 0.98 |
| 5 | Politics | 0.99 | 0.99 | 0.99 |
| 6 | law and order | 0.98 | 0.98 | 0.98 |
| 7 | Earthquake | 1.00 | 1.00 | 1.00 |
| 8 | Showbiz | 0.99 | 0.99 | 0.99 |
| 9 | Fraud and corruption | 0.99 | 0.99 | 0.99 |
| 10 | Rain/weather | 0.99 | 1.00 | 0.99 |
| 11 | Sexual Assault/Intercourse | 0.99 | 1.00 | 1.00 |
| 12 | Business | 1.00 | 0.95 | 0.97 |
| Overall accuracy | 98.96% | | | |

**Table 13 Decision Tree performance using the 'Title and Length'.**

| Label | Event | Precision | Recall | F1_Measure |
|---|---|---|---|---|
| 1 | Sports | 1.00 | 1.00 | 1.00 |
| 2 | Inflation | 1.00 | 1.00 | 1.00 |
| 3 | Murder and Death | 0.99 | 0.99 | 0.99 |
| 4 | Terrorist Attack | 0.99 | 0.99 | 0.99 |
| 5 | Politics | 1.00 | 1.00 | 1.00 |
| 6 | law and order | 0.99 | 1.00 | 0.99 |
| 7 | Earthquake | 1.00 | 1.00 | 1.00 |
| 8 | Showbiz | 1.00 | 0.99 | 1.00 |
| 9 | Fraud and corruption | 1.00 | 0.99 | 1.00 |
| 10 | Rain/weather | 1.00 | 1.00 | 1.00 |
| 11 | Sexual Assault/Intercourse | 1.00 | 1.00 | 1.00 |
| 12 | Business | 1.00 | 0.98 | 0.99 |
| Overall accuracy | 99.63% | | | |

The semantics of the script written in the Urdu language is quite different from that of English and Arabic Language which causes the low performance of SVM and k-NN as compared to Random Forest.

## DISCUSSION

Event extraction and classification are tightly coupled with processing resources *i.e.*, part-of-speech tagger (PoS), text annotators, and contextual insights. Meanwhile, the usage of local languages being highly preferred over social media is creating problems to analyze by existing tools. Urdu is one of those languages that have a considerable number of users and a huge bulk of data on social networks. It contains worthy insights that are

**Table 14 Random forest performance using the 'Title and Length'.**

| Label | Event | Precision | Recall | F1_Measure |
|---|---|---|---|---|
| 1 | Sports | 1.00 | 1.00 | 1.00 |
| 2 | Inflation | 1.00 | 1.00 | 1.00 |
| 3 | Murder and Death | 1.00 | 1.00 | 1.00 |
| 4 | Terrorist Attack | 1.00 | 0.99 | 1.00 |
| 5 | Politics | 1.00 | 1.00 | 1.00 |
| 6 | law and order | 1.00 | 1.00 | 1.00 |
| 7 | Earthquake | 1.00 | 1.00 | 1.00 |
| 8 | Showbiz | 1.00 | 1.00 | 1.00 |
| 9 | Fraud and corruption | 1.00 | 1.00 | 1.00 |
| 10 | Rain/weather | 1.00 | 1.00 | 1.00 |
| 11 | Sexual Assault/Intercourse | 1.00 | 1.00 | 1.00 |
| 12 | Business | 1.00 | 1.00 | 1.00 |
| Overall accuracy | 99.92% | | | |

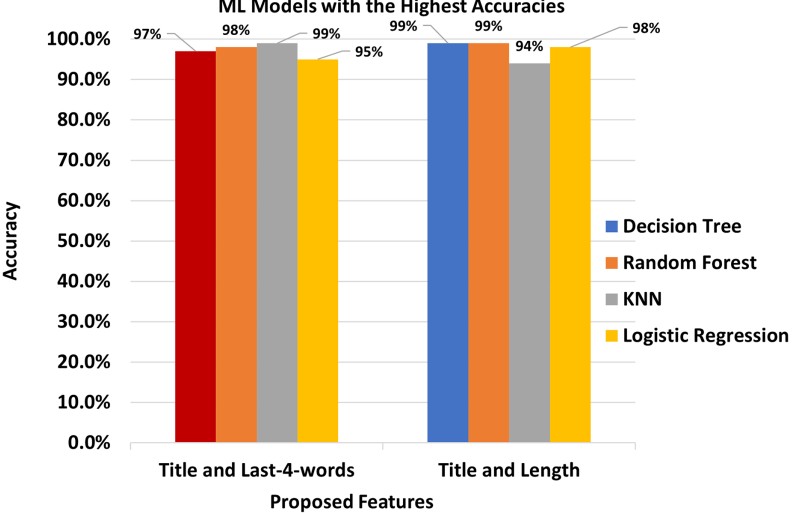

**Figure 3 The Best features and the best classifiers.**

necessary to process for different purposes like to improve security, to understand the intentions, trends, and mindset of people. We performed event classification written in Urdu language text on different social media like platforms. The evaluation of results is presented after analyzing multiple features *i.e.*, Length, Last-4-words, Title, and combination of all these features converged our findings to conclude that length and Last-4-words are basic features to classify multiclass events but showed 53% accuracy. To improve the accuracy of the proposed system, we integrated "Title" as the feature with other two features *i.e.*, Length and Last-4-words. The combination of "Title" with "Length and Last-4-words" improved the performance of the proposed system and showed the highest results as reported in the abstract.

As described in the dataset section that the dataset is imbalanced and contains multiple classes. To validate the accuracy of results not only TP, FP, TN, FP reported but also the standard performance evaluations parameters *i.e.*, precision, recall, and f1-measure are reported in Tables 11–14.

Furthermore, extracting and classification of events from resource-poor language is an interesting and challenging task. There are no standard (benchmark) datasets and word embedding models like Word2Vec or Glove (Exists for the English Language) for Urdu language text.

## CONCLUSIONS

A massive amount of Urdu textual data exists on social networks and news websites. Multiclass event classification for Urdu text at the sentence level is a challenging task because of the few numbers of words and limited contextual information.

The selection of appropriate features and approaches is necessary to classify multiclass events written in Urdu language text.

The deep analysis of the structure of sentences written in the Urdu language leads us to select these appropriate features *i.e.* title, length, the last four words, and a combination of all these features.

Experimental results showed that non-of single feature is capable to classify multiclass events. In contrast to the different combinations of these features *i.e.* title and last-4-words, title and length and last-4-words and length showed considerable results.

Count_ Vectorizer and TF-IDF feature generating techniques are used to convert text into (numeric) real value for machine learning models. Random Forest classification model showed 52% and 53% accuracy for last-4-words and combination of length and last-4-words.

The title is the key feature that can dramatically improve the performance of event classification models that works on a sentence level. Combining title with last-4-words and length showed the highest accuracies *i.e.* 98.00% and 99.00% for Random Forest and *k*-NN classifiers, respectively.

## FUTURE WORK

- In a comprehensive review of Urdu literature, we found a few numbers of referential works related to Urdu text processing. One of the main issues associated with the Urdu language research is the unavailability of the appropriate *corpus* like the data set of Urdu sentences representing the event; the close-domain PoS tagger; the lexicons, and the annotator, *etc.*
- There is a need to develop the supporting tools *i.e.*, the PoS tagger, the annotation tools, the dataset of the Urdu-based languages having information about some information associated with the events, and the lexicons can be created to extend the research areas in the Urdu language.

- In the future, many other types of events and other domains of information like medical events, social, local, and religious events can be classified using the extension of machine learning *i.e.*, deep learning.
- In the future grammatical, contextual, and lexical information can be used to categorize events. Temporal information related to events can be further utilized to classify an event as real and retrospective.
- Classification of events can be performed at the document level and phrase level.
- Deep learning classifiers can be used for a higher number of event classes.

## ACKNOWLEDGEMENTS

I am very grateful to the Chairman of the Faculty of Computing and Head of Department of Software Engineering of Islamia University Bahawalpur, Pakistan. They encouraged and guided me to make research work more interesting.

### Funding

The authors received no funding for this work.

### Competing Interests

The authors declare that they have no competing interests.

### Author Contributions

- Malik Daler Ali Awan conceived and designed the experiments, performed the experiments, analyzed the data, performed the computation work, prepared figures and/or tables, authored or reviewed drafts of the paper, and approved the final draft.
- Nadeem Iqbal Kajla performed the experiments, analyzed the data, performed the computation work, prepared figures and/or tables, authored or reviewed drafts of the paper, and approved the final draft.
- Amnah Firdous performed the experiments, analyzed the data, performed the computation work, prepared figures and/or tables, authored or reviewed drafts of the paper, and approved the final draft.
- Mujtaba Husnain conceived and designed the experiments, performed the experiments, analyzed the data, authored or reviewed drafts of the paper, and approved the final draft.
- Malik Muhammad Saad Missen conceived and designed the experiments, analyzed the data, authored or reviewed drafts of the paper, and approved the final draft.

### Data Availability

The input file and code are available in the Supplemental Files and at GitHub: https://github.com/unique-world/Social-Event-Classification-.

## Supplemental Information

Supplemental information for this article can be found online at http://dx.doi.org/10.7717/peerj-cs.775#supplemental-information.

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
