# Peer review of "Event classification from the Urdu language text on social media"

_PeerJ Computer Science, doi:10.7717/peerj-cs.775_

## Round 0.1 · original submission · Major Revisions

In order to be published you should address all issues identified by the reviewers. In particular it is very important to be clear about the nature of the work in the title and abstract, and to describe all methods, all algorithms used, and to be clear about the size of the training set.

·

Basic reporting

The title of the paper is “Event classification from text existing on social media”
it's not mentioned in the title and abstract, that the data is in Urdu language !!!

Experimental design

In the lines 182-183 the authors mentioned that “
In comparison, we used sentences of Urdu language for classification and explored the textual features of sentences. We disclosed all textual features in detail in this paper that were not reported ever in state-of-art.”
Comment :
The authors didn’t mention in details all the textual features for the Urdu language, and they didn't provide enough examples for each feature so the reader can understand it.


the authors didn't add a section for the machine learning algorithms they used in the paper, or even add a references for these algorithms!!!!


the authors didn't mentioned the stemmer or POS tool they used in the experiments.

the authors mentioned "The dataset consists of more than 1 million (1,02,962) labeled instances of twelve (12)
22 different types of events." what was the size of training data , and the size for testing data!!!!

the authors didn't include any example for their work!!!

Validity of the findings

the authors didn't experiment all choices of the extracted features, and they just did the experiment on "title and last 4 word" !!! only

the tables 7-10 not explained.
All the figures not explained.

the findings not clear and accurate!!!

Additional comments

NO comments

·

Basic reporting

The author works in the text classification domain and performs an experiment on the event classification in the Urdu language. The author used state of the art machine learning models and show that the KNN outperform all machine learning models. The author used a dataset contain 1 million records and 12 different labels taken from social me dia.

Experimental design

1-The author used TF-IDF and count vectorizer for features extraction. Why author used only TF-IDF and BoW count vectorizer? the author should use word2vec in comparison with TF-IDF and count vectorizer see the article "Rustam, F., Ashraf, I., Mehmood, A., Ullah, S. and Choi, G.S., 2019. Tweets classification on the basis of sentiments for US airline companies. Entropy, 21(11), p.1078."

2- Author should check the IDF formula again.
3- KNN is best performer when features set will be small. Find a supporting article for that where knn outperform on a large features set.
4- Author used state of the art method which have already lots of used in classification what is the novality of author work. work in specific domain is not a novality.

Validity of the findings

No comment

Additional comments

1- The experimental diagram should follow the full experimental flow.
2- author work on specific language how to deal with preprocessing of text there should be a more clear discussion about the library and self-generated library if anything used by the author.
3- the author should perform a comparison with results without preprocessing of text.
4- grammar should be checked thoroughly.

---

## Round 0.2 · Major Revisions

In order to be published, major revisions are required in particular
see Reviewer 2's comments about lack of detail in results.

In multiple sections, you say your dataset consists of more than one million sentences/instances. But the actual number seems to be 0.1 million. Also do not write "1,02,962". The comma is in the wrong place, and it misleads the reader into thinking there are one million, rather than one-tenth that. Write "102,962" instead.

You also do not report the number of distinct posts. A single post has
one title and many sentences. As reviewer 1 points out, you report
unreasonably high accuracy when using Title as a feature. I suspect
that when you split the dataset into training and test sets, you
are splitting by sentence and not by post. It is therefore not
surprising that you can predict with 100% accuracy based on the title!

If this is true you need to redo the analysis to ensure that the
features of the post do not end up polluting your results.

I think your reporting of the results needs improving. Figures 4-6 are
redundant with the tables. You should combine these into a smaller
number of tables or figures. But this point is secondary if the
evaluation is flawed.

I also ask that you improve your literature review section. In particular, you say:

> Urdu textual contents explored [27] for classification using the majority voting algorithm. They categorized Urdu text into seven classes i.e., Health, Business, Entertainment, Science, Culture, Sports, and Wired. They used 21769 news documents for classification and reported 94% precision and recall. Dataset evaluated using these algorithms, Linear SGD, Bernoulli Naïve Bayes, Linear SVM, Naïve Bayes, random forest classifier, and Multinomial Naïve Bayes. Textual classification is close to our problem that is events classification by text at the sentence level, but it is completely different. They did not report the overall accuracy of the system for multiple classes. The information about feature selection is also omitted by the researchers but comparatively, we disclosed the feature selection, engineering, and accuracy of classifiers for multi-classes. Our dataset set consists of 1,02,960 instances of sentences and twelve (12) classes that are comparatively very greater

Citation [27] (Daud et al) is a survey, summarizing the current state
of the art. When you say "they reported 94% precision and recall" and
"They did not report the overall accuracy of the system" this is
misleading. Daud is summarizing the results of Sajjad and Schmid
(2009) (amongst many others). It is not appropriate for you to
criticize them for not reporting all statistics, as it is a review
paper.

I recommend that you take this section out, and instead cite the
primary research in the papers cited in Daud et al, and compare your
work against the primary research works.

·

Basic reporting

no comment

Experimental design

no comment

Validity of the findings

The accuracy for the ML algorithms with the features , are not reasonable , increasing from 17% to 85% , then the final results 99% !!!
The authors didn't mention any weaknesses of their model.

In many English and Arabic researches , the SVM and KNN , were the best ML algorithms , but in your research they are 's not, can the authors mention the reasons!!

Additional comments

The examples should be written in Urdu , and English language as well,

·

Basic reporting

Comments are below

Experimental design

Comments are below

Validity of the findings

Comments are below

Additional comments

The author asks to revise the manuscript to improve the quality of the article and for this, I have mentioned some comments for the author. Lots of comments have cooperated but still, there are some weak areas in the manuscript.
1- Abstract's first 3 and 4 lines are too general and the overall abstract should be more attractive. best performers' results should be added to the abstract.

2- Results section should contain more detail about the results and represent the significance of the models.

---

## Round 0.3 · Major Revisions

Thank you for your latest version.

Unfortunately, you have not adequately addressed the critiques. Please note that when submitting a rebuttal to a critique from editors or reviewers it is not acceptable to say "All the recommendation are incorporated in the article." and provide the track changes version. You should instead offer a point by point response to all concerns raised, highlighting which sections were added or taken out. This is particularly important if the reviewer or editors have raised serious concerns.

It is also problematic that the whole analysis seems to have changed, some of the tables in this new version appear to be different, there is no explanation of the changes.

In this case it appears you have only partially addressed the concerns raised. You have addressed minor issues by subtracting text but you ignored or made minor efforts to address the major issues. Please read my summary from the previous round, and address each concern in detail.

Frankly, some of the changes raise suspicions. Previously a reviewer had concerns that using Title alone had unreasonable accuracy, and I gave you a few reasons why this might be the case. The correct thing to do here was to address these concerns. Instead you seem to have simply dropped this table in the results.

---

## Round 0.4 · Minor Revisions

I took over handling your manuscript as the previous Academic Editor became unavailable.

While the content of the paper is deemed scientifically correct, the writing needs to improve that the contributions and the findings are clearly stated throughout, particularly in the abstract, introduction and conclusion. Please revise these sections to make this clear.

Please proofread the paper to improve its readability.

·

Basic reporting

No comments

Experimental design

No comments

Validity of the findings

no comments

Additional comments

no comments

·

Basic reporting

The author presented the event classification for the Urdu language text existing on social media and news channels. The dataset contains more than 0.1 million (102,962) labeled instances of twelve (12) different types of events. Title, Length, and last-4-words of a sentence are used as features to classify events. The Term Frequency-Inverse Document Frequency (tf-idf) showed the best results as a feature vector to evaluate the performance of the six popular machine learning classifiers. The author resolves the comments but still lots of things should be improved especially the abstract. such as:

These mention two sentences are the same in the abstract why? Random Forest (RF), Decision Tree, and k-Nearest Neighbor outperformed among the other classifiers. Random Forest and K-Nearest Neighbor are the classifiers that out-performed among other classifiers by achieving 98.00% and 99.00% accuracy, respectively.

The Abstract didn't contain detail about contribution and methodology. it's too general and short.

Experimental design

No commets

Validity of the findings

The dataset is imbalanced then the results are too significant justifies that the models are no overfitted on majority class data.

The author gives the confusion matrix report in terms of TP, TN, FP, and FP.. These terms are useful for binary classification how they adjust them in multiclass classification. add some visual infromation.

Additional comments

No comemnt

---

## Round 0.5 · accepted · Accept

The current revision addresses the minor revisions suggested in the final round and it can now be accepted for publication in its current form.